# Distribution Patterns and Assembly Mechanisms of Rhizosphere Soil Microbial Communities in *Schisandra sphenanthera* Across Altitudinal Gradients

**DOI:** 10.3390/biology14080944

**Published:** 2025-07-27

**Authors:** Weimin Li, Luyao Yang, Xiaofeng Cong, Zhuxin Mao, Yafu Zhou

**Affiliations:** Xi’an Botanical Garden of Shaanxi Province, Institute of Botany of Shaanxi Province, Xi’an 710061, China; jake219@163.com (W.L.); yangluyao@xab.ac.cn (L.Y.); congxiaofeng312@xab.ac.cn (X.C.); zhuxinmao@gmail.com (Z.M.)

**Keywords:** *Schisandra sphenanthera*, altitude, soil microorganisms, community assembly, co-occurrence network

## Abstract

*Schisandra sphenanthera* is a traditional medicinal plant that grows in mountainous regions. However, little is known about how the tiny organisms living in its root soil—known as microbes—change as the altitude increases. In this study, we collected soil samples from around the roots of this plant at four different elevations and analyzed the types and relationships of soil bacteria and fungi. We found that different elevations influenced which types of microbes were present and how they interacted. At higher altitudes, some microbes became more cooperative, while others became more competitive. The nutrients in the soil, as well as plant growth and altitude, played key roles in shaping these microbial communities. These changes also affected how soil carbon, an important element for soil health and climate balance, is stored and cycled. Our findings help explain how soil life adjusts to mountain environments and provide useful information for protecting both this valuable plant and the surrounding ecosystem.

## 1. Introduction

*Schisandra sphenanthera* (commonly known as southern magnolia vine), a perennial woody liana, belongs to the genus *Schisandra* within the family *Schisandraceae*. This species is widely distributed across China, including in the northwestern, central, eastern, and southwestern regions, and holds significant value as a traditional Chinese medicinal plant. Its fruits are extensively utilized in traditional medicine for their astringent and consolidating properties, as well as for their efficacy in tonifying qi, promoting fluid production, nourishing the kidneys, and calming the mind. Due to its high clinical demand, *S. sphenanthera* is of substantial pharmacological importance [1,2]. Schisandrin A and Schisandrin B, the representative lignan compounds derived from plants of the Schisandra genus, exhibit complementary and synergistic pharmacological activities through multi-target regulatory mechanisms [3]. In the domain of hepatoprotection, both compounds demonstrate significant inhibitory effects against liver injury. Regarding neuroprotection, Schisandrin B, due to its lipophilic nature, efficiently crosses the blood–brain barrier, whereas Schisandrin A enhances dopaminergic neuron survival by modulating the GABAergic system and inhibiting monoamine oxidase B (MAO-B) activity [4]. Furthermore, both compounds exhibit notable antitumor properties. Schisandrin B promotes gut health by increasing the abundance of beneficial bacteria such as Bifidobacterium and enhancing intestinal barrier integrity, while Schisandrin A attenuates intestinal inflammation via suppression of the TLR4/MyD88 signaling pathway. These synergistic and complementary mechanisms underpin the therapeutic potential of these compounds in liver protection and the management of neurodegenerative diseases [5].

The Qinling Mountains, a critical climatic divide in China, exhibit pronounced environmental contrasts between their southern and northern slopes. The southern slope experiences warm and humid conditions, whereas the northern slope is characterized by relatively dry and cold climates. These climatic disparities significantly influence the growth and fruit traits of *S. sphenanthera.* Studies indicate that altitude, mean annual temperature, and geographic coordinates (longitude and latitude) are key factors driving phenotypic variation in its fruits [6,7]. *S. sphenanthera* is abundantly distributed across both slopes of the Qinling Mountains, typically thriving at elevations ranging from 600 to 3000 m. In this region, the species predominantly inhabits forest edges and shrublands, where favorable growth conditions—such as adequate sunlight, suitable temperature, and humidity—support its vegetative development and fruit maturation [8].

Altitude is a critical factor in selecting cultivation areas for *Schisandra sphenanthera*, as elevation gradients significantly alter environmental conditions such as light, temperature, moisture, and soil resources, thereby influencing soil physicochemical properties [9]. In mountain ecosystems, vegetation and soil microbial community characteristics undergo pronounced shifts with increasing elevation [10]. Soil microorganisms, being integral components of terrestrial ecosystems, exhibit high sensitivity to environmental changes, and their community structure and diversity are closely linked to soil nutrient transformation [11,12]. Therefore, investigating the structure and composition of soil microbial communities is essential for optimizing the soil microenvironment, sustaining biodiversity, and maintaining ecological equilibrium. Recent advancements in sequencing technologies have deepened our understanding of altitudinal patterns in soil microbial communities. For instance, Murugan et al. observed higher fungal abundance in high-altitude soils compared with low-altitude regions in the Alps [13]. Conversely, Wang et al. reported a decline in soil microbial diversity with increasing elevation on Fanjing Mountain [14]. Ren et al. [15], studying Taibai Mountain, found that bacterial α-diversity peaked at mid-elevations, while fungal α-diversity remained unaffected by altitude. Additionally, elevation exerted a stronger influence on bacterial β-diversity than on fungal β-diversity. Specifically, the relative abundances of *Acidobacteriota* and *Actinobacteriota* decreased significantly with rising elevation, whereas those of *Chloroflexi* and *Nitrospirae* were more prevalent at mid-elevations.

Soil microorganisms play a pivotal role in terrestrial ecosystems by regulating organic matter cycling processes [16]. The application of high-throughput sequencing in microbiome research has significantly advanced our understanding of soil microbial communities and their functions, particularly in elucidating rhizosphere microbial diversity, community assembly, and functional traits linked to plant nutrient acquisition mechanisms [17]. The maturation stage of *Schisandra sphenanthera* is critical for determining fruit quality, during which the rhizosphere microbial community structure serves as a key influencing factor. Therefore, investigating microbial community characteristics under varying altitudinal conditions is essential for understanding the plant’s growth, development, and metabolic processes. This study hypothesizes that altitudinal variation significantly alters the rhizosphere soil microbial diversity of *Schisandra sphenanthera*, leading to pronounced shifts in both dominant and rare microbial taxa. These changes may be attributed to decreasing temperatures with increasing elevation (approximately 0.6 °C per 100 m), which likely suppress microbial metabolic activity. To test this hypothesis, we employed high-throughput sequencing and quantitative fluorescence PCR to analyze the microbial community structure and diversity across different elevations. Molecular ecological network analysis was utilized to explore intergeneric microbial interactions. Furthermore, a null model was applied to decipher microbial community assembly processes and to identify key environmental drivers. These findings aim to provide a theoretical foundation for regulating soil microecology in *S. sphenanthera* habitats along altitudinal gradients.

## 2. Materials and Methods

### 2.1. Study Area

Zhashui County (33°25′–33°56′ N, 108°50′–109°36′ E), located in Shangluo City, Shaanxi Province, lies 112 km south of Xi’an City on the southern slope of the Qinling Mountains. The county spans 72.7 km east–west and 55.2 km north–south, covering a total area of 2366.67 km^2^. Situated in the inland region of northwestern China, Zhashui exhibits climatic characteristics of both northern and southern zones: its northern part belongs to the warm temperate zone, while the southeastern area falls within the northern subtropical zone. The entire county is characterized by a transitional monsoon climate with distinct seasons, mild temperatures, and abundant rainfall. The study area is characterized by a warm temperate and semi-humid monsoon climate, with an average annual temperature ranging from 2 to 10 °C and annual precipitation between 850 and 950 mm. The predominant soil type is mountainous brown soil. This unique natural environment establishes Zhashui as an optimal habitat for diverse medicinal plants, earning it the title of “Natural Medicine Treasury”. The region’s ecological conditions—particularly its climate and resource richness—provide an ideal environment for wild *Schisandra sphenanthera*. Zhashui is recognized as a concentrated distribution area and an authentic production area for *S. sphenanthera*, as validated through historical herbal studies. Huanghualing Mountain in Yingpan Town, Zhashui County, has been identified as the core distribution area for wild *S. sphenanthera* germplasm resources, with abundant reserves of superior quality.

### 2.2. Sample Collection

Sampling was conducted on 6 July 2024, at Huanghualing Mountain in Yingpan Town, Zhashui County, on the southern slope of the Qinling Mountains. Four altitudinal gradients were selected: 900 m (HB1), 1100 m (HB2), 1300 m (HB3), and 1500 m (HB4). Three independent biological replicates were collected at each altitude gradient, totaling 12 composite rhizosphere soil samples (3 replicates × 4 altitudes). For each replicate, five healthy *Schisandra sphenanthera* plants were selected. Surface litter, dead branches, and plant debris were carefully removed, and soil from the root zone (0–20 cm depth) was collected using a sterilized soil auger. These samples were immediately placed in sterile polyethylene bags. The collected soil was homogenized and divided into two subsamples: one was stored at −80 °C and sent to Shanghai Majorbio Bio-pharm Technology Co., Ltd. (Shanghai, China) for microbial analysis, while the other was air-dried, ground, and sieved (2 mm mesh) for physicochemical property determination, including pH, total carbon (TC), total nitrogen (TN), total phosphorus (TP), and total potassium (TK). Additionally, plant tissues from each elevation were collected to analyze nutrient contents (N, P, K).

### 2.3. Soil Physicochemical Analysis

Soil pH was measured using a pH meter (Mettler Toledo, Greifensee, Switzerland) at a soil-to-water ratio of 1:2.5 (*w*/*v*). Soil organic carbon (SOC) was determined via the potassium dichromate external heating method [18]. Total nitrogen (TN) was quantified using the semi-micro Kjeldahl method [19]. Total phosphorus (TP) was analyzed by ammonium molybdate spectrophotometry [20], and total potassium (TK) was measured via flame photometry [21]. The concentrations of schisandrin A and schisandrin B in *Schisandra sphenanthera* were determined using high-performance liquid chromatography (HPLC) equipped with either an ultraviolet (UV) detector or a diode array detector (DAD) due to the strong UV absorbance of schisandrins near 254 nm. Chromatographic conditions were as follows: a mobile-phase flow rate of 1.0 mL/min, a column temperature of 30 °C, and a detection wavelength of 254 nm [22].

### 2.4. Soil DNA Extraction, High-Throughput Sequencing, and Bioinformatics Analysis

Total DNA was extracted from 0.25 g of moist soil using the PowerSoil^®^ DNA Isolation Kit (MO BIO Laboratories, Carlsbad, CA, USA) following the manufacturer’s protocol. To minimize experimental variability, triplicate extractions were pooled for each sample. DNA extracts were purified by electrophoresis on a 1% agarose gel, and their concentration and purity were assessed using a NanoDrop UV-Vis spectrophotometer (ND-2000c, NanoDrop Technologies, Wilmington, DE, USA). Bacterial 16S rRNA gene amplification was performed with primers 515F (5′-GTGCCAGCMGCCGCGGTAA-3′) and 907R (5′-CCGTCAATTCMTTTRAGTTT-3′), while fungal ITS regions were amplified using primers ITS5-1737F (5′-GGAAGTAAAAGTCGTAACAAGG-3′) and ITS2-2043R (5′-GCTGCGTTCTTCATCGATGC-3′) [23]. PCR products were sequenced on the Illumina MiSeq platform (PE300) by Shanghai Majorbio Bio-pharm Technology Co., Ltd. (Shanghai, China). Raw sequences were processed using the DADA2 package [24] to generate high-resolution amplicon sequence variants (ASVs). Low-quality reads were trimmed using cutadapt [25], followed by error-rate estimation via the learnErrors function in DADA2. Overlapping regions of paired-end reads were merged using the mergePairs algorithm. Chimeric sequences were identified and removed using DADA2’s built-in chimera detection module, resulting in refined ASVs. Taxonomic annotation of ASVs was performed by aligning representative sequences against the SILVA 138 (for bacteria) and UNITE 8.0 (for fungi) databases with a confidence threshold of 80%. The α-diversity indices (Chao1, Ace, Shannon, and Simpson) of soil microbial communities across altitudinal gradients were calculated at the OTU level using the “vegan” and “picante” packages in R. These indices reflect the species richness (Chao1, Ace) and evenness (Shannon, Simpson) within each sample. The β-diversity was assessed using the Bray–Curtis dissimilarity index, computed via the vegdist function in the “vegan” package. This metric quantifies compositional differences in microbial communities between samples along the elevation gradient.

### 2.5. Data Processing

Soil physicochemical properties and microbial community composition data were processed using SPSS 26.0 and Excel 2010. One-way analysis of variance (ANOVA) and multiple comparisons (LSD method; *p* = 0.05) were used to analyze significant differences. The composition and diversity of microbial communities were analyzed using the Sanger cloud platform provided by the Majorbio Cloud Platform (https://cloud.majorbio.com/, accessed on 11 February 2025). Operational Taxonomic Units (OTUs) with a Spearman correlation coefficient of R > 0.5 and a significance level of *p* < 0.05 among soil microbial genera were selected. The “igraph” package in the R software (version 1.6.0.) was used to construct microbial community correlation networks, and Gephi 9.2 software was employed for network visualization analysis. The “mntd” and “ses.mntd” functions in the “picante” package of the R software were used to calculate the Beta Nearest Taxon Index (βNTI). When |βNTI| > 2, it indicates that the community assembly process is deterministic. When |βNTI| < 2, the community assembly is mainly governed by stochastic processes [26]. The “vegan” package in the R software was used to determine the RCbray values. Homogeneous selection, dispersal limitation, and undominated processes are represented by RCbray < −0.95, RCbray > 0.95, and RCbray < 0.95, respectively. Redundancy analysis (RDA) of soil microbes and environmental factors was conducted using CANOCO 5.0.

## 3. Results and Analysis

### 3.1. Soil Physicochemical Properties

As shown in Figure 1, significant differences in soil physicochemical properties were observed across altitudinal gradients. The HB4 elevation exhibited significantly higher soil pH, total potassium (TK), plant total nitrogen (TN), and plant total potassium (TK) compared with other treatments (*p* < 0.05). In contrast, HB3 showed the highest values for soil total carbon (TC), soil TN, soil total phosphorus (TP), plant TP, and the concentrations of schisandrin A and schisandrin B (*p* < 0.05). Compared with HB1, the HB4 treatment resulted in increases of 2.94% in soil pH, 13.42% in total soil potassium, 45.67% in total plant nitrogen, and 4.44% in total plant potassium. In the HB3 treatment, total soil phosphorus and total plant phosphorus increased by 66.90% and 89.59%, respectively, while levels of total soil carbon, total soil nitrogen, schisandrin A, and schisandrin B all more than doubled. Notably, no significant differences were detected in plant TC among all the treatments.

### 3.2. Soil Microbial Community Composition

High-throughput sequencing revealed distinct microbial community structures across altitudinal gradients (Figure 2). Bacterial communities were dominated by *Proteobacteria*, *Acidobacteriota*, *Actinobacteriota*, and *Chloroflexi*, with total relative abundances of 79.19% (HB1), 75.99% (HB2), 77.94% (HB3), and 76.54% (HB4), respectively, indicating a decline in dominant bacterial phyla with increasing elevation. Specifically, the relative abundance of *Proteobacteria* was lowest at HB1 (20.72%) and peaked at HB3 (31.03%), whereas *Acidobacteriota* exhibited the highest abundance at HB1 (29.18%) and the lowest at HB3 (21.41%). *Patescibacteria*, *Latescibacterota*, *Entotheonellaeota*, *Desulfobacterota*, and *Bdellovibrionota* were identified as rare bacterial phyla, with relative abundances of 0.79%, 0.58%, 0.54%, 0.50%, and 0.38%, respectively. The relative abundance of *Patescibacteria* was lowest in the HB3 treatment (0.69%) and highest in HB2 (0.98%). The relative abundances of *Latescibacterota*, *Entotheonellaeota*, and *Desulfobacterota* were highest in the HB4 treatment, reaching 0.73%, 0.65%, and 0.62%, respectively.

For fungal communities, *Ascomycota* and *Basidiomycota* were the dominant phyla, with total relative abundances of 57.38% (HB1), 88.01% (HB2), 79.40% (HB3), and 75.07% (HB4), suggesting an increase in dominant fungal phyla at higher elevations. *Ascomycota* showed the lowest abundance at HB1 (45.04%) and the highest at HB2 (73.07%), while *Basidiomycota* reached its maximum at HB4 (15.78%) and minimum at HB3 (8.80%). *Basidiobolomycota*, *Olpidiomycota*, *Kickxellomycota*, *Zoopagomycota*, *Aphelidiomycota*, *Monoblepharomycota*, and *Blastocladiomycota* were identified as rare fungal phyla, with relative abundances of 0.08%, 0.04%, 0.04%, 0.02%, 0.02%, 0.01%, and 0.0005%, respectively. The relative abundances of *Olpidiomycota*, *Kickxellomycota*, *Aphelidiomycota*, and *Blastocladiomycota* were highest in the HB3 treatment, reaching 0.13%, 0.09%, 0.04%, and 0.002%, respectively. In contrast, *Basidiobolomycota* and *Monoblepharomycota* exhibited their highest relative abundances in the HB4 treatment at 0.30% and 0.03%, respectively.

Principal coordinate analysis (PCA) based on the Bray-Curtis dissimilarity index was applied to assess structural shifts in soil bacterial and fungal communities across altitudinal gradients (Figure 3). For bacterial communities, PCA separated the samples into three distinct clusters: (1) HB1 and HB2, (2) HB3, and (3) HB4. Notably, HB4 exhibited the farthest separation from other groups, indicating pronounced divergence in the bacterial community structure at higher elevations. Similarly, fungal communities clustered into three groups: (1) HB2 and HB4, (2) HB1, and (3) HB3, with HB3 showing the greatest dissimilarity with other treatments. Analysis of similarities (ANOSIM) further confirmed that altitude significantly influenced both bacterial and fungal community compositions (*p* < 0.05).

### 3.3. Soil Microbial Community Diversity

The α-diversity indices of soil microbial communities are shown in Figure 4. For bacterial communities, the Chao1, Ace, and Shannon indices decreased with increasing elevation, while the Simpson index increased. Notably, HB2 exhibited the highest bacterial Chao1, Ace, and Shannon indices but the lowest Simpson index, whereas HB4 displayed the opposite trend. In contrast, fungal Chao1 and Ace indices initially increased and then declined with elevation, while the Shannon index increased and the Simpson index decreased. Specifically, HB3 showed the highest fungal Chao1 and Ace indices, whereas HB4 had the highest Shannon index and the lowest Simpson index.

### 3.4. Soil Microbial Community Assembly Mechanisms

To elucidate the drivers of altitudinal variation in *Schisandra sphenanthera* rhizosphere microbial communities, a null model was employed to analyze the intrinsic assembly processes (Figure 5a–c). For bacterial communities, deterministic processes dominated across all elevations, primarily driven by heterogeneous selection. The contribution of heterogeneous selection and undominated processes declined with increasing elevation, while dispersal limitation increased. In contrast, fungal community assembly was governed by stochastic processes (Figure 5d–f), with dispersal limitation (a stochastic process) dominating at lower elevations but decreasing at higher altitudes, whereas undominated processes became more prevalent.

Further correlation analysis (Figure 6) revealed that bacterial βNTI was significantly negatively correlated with soil total phosphorus (TP) (*p* < 0.05), while fungal βNTI showed significant positive correlations with soil TP and plant total potassium (TK) (*p* < 0.05). Additionally, bacterial Shannon diversity was negatively correlated with soil TP, soil TK, plant TN, and plant TK (*p* < 0.05), whereas fungal Shannon diversity exhibited positive correlations with soil TP, plant TN, plant TP, and the concentrations of schisandrin A and B (*p* < 0.05).

### 3.5. Co-Occurrence Network Characteristics of Soil Microbial Communities

Significant differences were observed in soil microbial co-occurrence network characteristics across different altitudes (Figure 7; Table 1). For bacterial co-occurrence networks, the HB2 treatment exhibited the highest number of edges (185), nodes (2591), average weighted degree, and network density, indicating a higher complexity of the bacterial co-occurrence network under this treatment, with more intricate interactions among species. However, it showed lower modularity, suggesting the absence of clearly defined substructures. In addition, the HB4 treatment had a higher proportion of positive correlations and a lower proportion of negative correlations compared with other treatments, while the HB1 treatment showed the opposite trend, with a lower proportion of positive correlations and a higher proportion of negative correlations. These results suggest that bacterial communities at higher altitudes tend to exhibit stronger cooperative interactions among species, whereas those at lower altitudes display stronger competitive relationships. In terms of fungal co-occurrence network characteristics, the HB4 treatment showed the highest number of edges (160), nodes (1917), average weighted degree, and network density, indicating a more complex fungal co-occurrence network and more intricate species interactions. Similarly, the network had low modularity, lacking distinct substructures. Furthermore, the HB2 treatment exhibited a higher proportion of positive correlations and a lower proportion of negative correlations, whereas the HB4 treatment had a lower proportion of positive correlations and a higher proportion of negative correlations. This implies that fungal communities at lower altitudes tend to have stronger cooperative interactions, while those at higher altitudes are characterized by more competitive relationships.

Further analysis of bacterial–fungal interaction networks revealed that under the HB4 treatment, the co-occurrence network composed of bacteria and fungi had the highest number of edges (307), nodes (7659), and average weighted degree, indicating a highly complex and interwoven network structure with strong interactions. Moreover, this network displayed higher modularity, with clearly defined substructures. Notably, the HB2 treatment had a higher proportion of positive correlations and a lower proportion of negative correlations than other treatments, suggesting that bacterial–fungal communities at lower altitudes are characterized by stronger cooperative interactions.

### 3.6. Relationships Between Soil Microbial Communities and Physicochemical Factors

To identify the dominant environmental factors influencing soil bacterial and fungal community compositions, redundancy analysis (RDA) was performed using bacterial and fungal phyla as response variables and soil physicochemical parameters as explanatory variables (Figure 8a,b). RDA results for the bacterial community showed that *Proteobacteria*, *Desulfobacterota*, *Patescibacteria*, *Firmicutes*, *Bacteroidota*, and *Nitrospirota* were positively correlated with STN, STC, PTP, STP, SA, and SB, while *Planctomycetota* was positively correlated with pH. For the fungal community, *Basidiomycota* showed a positive correlation with pH, while *Basidiobolomycota*, *Monoblepharomycota*, and *Rozellomycota* were positively correlated with PTC, PTN, and STK. *Chytridiomycota*, *Mucoromycota*, *Ascomycota*, and *Aphelidiomycota* were positively correlated with SB, PTK, SA, STP, and PTP. Additionally, *Olpidiomycota*, *Mortierellomycota*, *Zoopagomycota*, and *Blastocladiomycota* were positively correlated with STC and STN.

Further analysis using a structural equation model (SEM) (Figure 8c) revealed that elevation was significantly negatively correlated with soil nutrients and bacterial diversity but positively correlated with plant nutrients (*p* < 0.05). Both soil and plant nutrients showed significant positive correlations with bacterial diversity, while plant nutrients were significantly positively correlated with fungal diversity (*p* < 0.05). In terms of soil carbon sequestration, elevation and soil nutrients were significantly negatively correlated with soil carbon, whereas plant nutrients and fungal diversity were significantly positively correlated with soil carbon (*p* < 0.05).

## 4. Discussion

### 4.1. Effects of Altitude on Soil Microbial Community Characteristics in Schisandra sphenanthera

This study investigated the impacts of altitudinal gradients on soil bacterial and fungal communities in *Schisandra sphenanthera* rhizospheres across the southern Qinling Mountains, Shaanxi Province, using high-throughput sequencing to identify shifts in the dominant microbial phyla. Bacterial communities were dominated by *Proteobacteria*, *Acidobacteriota*, *Actinobacteriota*, and *Chloroflexi*. The relative abundance of *Proteobacteria* increased with elevation, whereas that of *Acidobacteriota* and *Actinobacteriota* declined. This trend may reflect Proteobacteria’s adaptability to hypoxic conditions at higher altitudes, while *Acidobacteriota* and *Actinobacteriota*—associated with soil health and organic matter decomposition—likely favor fertile soils at lower elevations [27,28]. The distribution of rare bacterial phyla varied across different elevation gradients. *Patescibacteria* exhibited the highest relative abundance at the HB2 site (0.98%) and the lowest at HB3 (0.69%), suggesting a preference for lower elevation environments, potentially due to higher organic matter availability and mildly hypoxic conditions in that region [29]. In contrast, *Latescibacterota*, *Entotheonellaeota*, and *Desulfobacterota* reached peak abundances at the highest elevation (HB4), indicating their adaptive capacity toward extreme high-altitude environments. The dominance of *Desulfobacterota* at high elevation may be attributed to its sulfate-reducing capabilities, which are advantageous under cold, low-oxygen conditions. Meanwhile, *Entotheonellaeota* may support host survival under high-altitude stress through symbiotic interactions [30,31].

For fungi, *Ascomycota* and *Basidiomycota* were dominant. *Ascomycota*, primarily saprophytic fungi, drive litter decomposition and organic matter degradation. Their numerical dominance in this study may stem from prolific asexual reproduction via conidia [32]. Both *Ascomycota* and *Basidiomycota* increased with elevation, alongside *Rozellomycota.* These trends align with their adaptations to cooler environments at higher altitudes. *Rozellomycota*, commonly found in polar and boreal regions, may gain competitive advantages under low-temperature, high-humidity conditions at elevated sites [33].

The elevational distribution patterns of rare fungal phyla reflect their distinct ecological adaptations. *Olpidiomycota*, *Kickxellomycota*, *Aphelidiomycota*, and *Blastocladiomycota* exhibited their highest relative abundances at the HB3 elevation, suggesting that these facultative or obligate parasitic fungi may rely on the specific host availability and relatively mild environmental conditions associated with this mid-elevation zone [34,35]. The enrichment of *Basidiobolomycota* and *Monoblepharomycota* as the elevation increased toward HB4 indicates specialized adaptive strategies for coping with extreme environments, potentially including cold-temperature metabolic adjustments and ultraviolet radiation defense mechanisms [36]. This vertical differentiation not only highlights the divergent responses of fungal functional groups to altitudinal variation but also underscores key processes underlying the maintenance of microbial diversity in mountainous ecosystems.

Soil microorganisms play a pivotal role in promoting plant growth. Among bacteria, members of *Proteobacteria*, such as rhizobia, contribute directly to plant nutrient acquisition through nitrogen fixation, phosphate solubilization, and auxin secretion. *Actinobacteriota* enhance plant development by producing antibiotics and phytohormones, thereby suppressing pathogens and modulating growth. *Chloroflexi*, through their involvement in photosynthesis and sulfur cycling, contribute to the optimization of the rhizosphere microenvironment [37,38]. In terms of fungi, *Ascomycota* (e.g., *Trichoderma*) secrete hydrolytic enzymes that degrade lignin and induce systemic resistance, while *Basidiomycota* expand the root absorption zone via ectomycorrhizal associations; their hyphal networks further facilitate carbon and nitrogen cycling [39]. Collectively, these microbial communities form an integrated multi-tiered pathway—encompassing root development, nutrient uptake, stress resilience, and pathogen suppression—that holistically enhances plant growth.

### 4.2. Effects of Altitude on Soil Carbon Sequestration in Schisandra sphenanthera

Altitude and soil nutrients exhibited significant negative correlations with soil carbon content, whereas plant nutrients and fungal diversity showed positive correlations. This pattern may arise from altitudinal climatic shifts: declining temperatures and reduced precipitation at higher elevations alter microbial activity and community structure, thereby modulating carbon decomposition and sequestration [40]. Lower temperatures suppress microbial metabolic rates, slowing organic matter decomposition and promoting carbon accumulation [41]. However, the decrease in precipitation can lead to soil drought, which negatively affects microbial growth and metabolism, thereby reducing soil carbon sequestration [42]. Additionally, elevation is often associated with the distribution of soil nutrients; soils at higher altitudes are typically more nutrient-poor, which can limit microbial activity and alter the community structure, ultimately influencing carbon sequestration [43]. The effect of soil nutrients on carbon sequestration mainly manifests through changes in nutrient availability, which directly affect microbial community composition and functional activity [44]. Essential nutrients such as nitrogen (N), phosphorus (P), and potassium (K) are required for microbial growth and metabolism. When nitrogen is abundant, microbial growth and metabolic activity are enhanced, leading to the accelerated decomposition of soil carbon, which is unfavorable for carbon sequestration. In contrast, under nitrogen-limited conditions, microbes may alter their metabolic pathways and reduce carbon decomposition, thereby enhancing soil carbon retention [45]. Similar effects are observed with phosphorus and potassium, whose availability can also shape microbial communities and influence soil carbon dynamics. The impact of plant nutrients on soil carbon sequestration is mainly reflected in the quantity and quality of plant residues. Through photosynthesis, plants absorb CO_2_ and convert it into biomass [46]. Plant residues—including litterfall and root detritus—serve as primary sources of soil organic matter. Higher nutrient availability promotes vigorous plant growth and greater biomass accumulation, leading to increased inputs in both the quantity and quality of residues, which favors soil carbon sequestration [47,48]. Fungal diversity also plays a key role in carbon sequestration due to fungi’s essential role in soil carbon cycling. Fungi decompose complex organic compounds into smaller, more accessible molecules for microbial uptake [49]. A higher fungal diversity reflects greater complexity and stability in microbial communities, enhancing ecosystem functions that contribute to carbon storage [50]. Furthermore, many fungi form symbiotic associations with plant roots (mycorrhizae), which improve plant nutrient uptake and promote plant growth and biomass accumulation, thereby contributing positively to soil carbon sequestration [51]. In summary, altitude-driven changes in temperature, precipitation, soil nutrients, plant productivity, and fungal diversity collectively regulate soil carbon dynamics through microbial- and plant-mediated pathways. Understanding these interactions provides a foundation for optimizing soil management in alpine ecosystems.

In summary, altitude-driven changes in temperature, precipitation, soil nutrients, plant productivity, and fungal diversity collectively regulate soil carbon dynamics through microbial- and plant-mediated pathways. Understanding these interactions provides a foundation for optimizing soil management in alpine ecosystems.

### 4.3. Effects of Altitude on Soil Microbial Interactions in Schisandra sphenanthera

Molecular ecological networks not only reflect interactions among different taxa within a community but also serve as effective tools for evaluating community complexity. They have been successfully applied to assess the effects of environmental factors on microbial communities [52]. In this study, soil microbial co-occurrence networks varied significantly across different elevations. The results indicated that bacterial communities at high elevations exhibited stronger cooperative relationships among species, whereas bacterial communities at low elevations were characterized by more competitive interactions. In contrast, fungal communities at low elevations tended to show stronger cooperation, while fungal communities at high elevations exhibited more competition.

These differences are likely attributed to harsher environmental conditions at higher elevations, such as lower temperatures and reduced moisture availability [53]. Such conditions impose greater survival stress on microbial communities, prompting species within bacterial and fungal communities to strengthen cooperative interactions in order to enhance their competitive ability. For example, bacterial species may engage in syntrophic interactions or share resources through the production of antibiotics or complementary metabolic pathways, thus improving the resilience of the entire community [54]. Similarly, fungal species may cooperate through mycelial networks, facilitating resource sharing and enhancing their overall survival capacity [55]. Conversely, environmental conditions at lower elevations are relatively more favorable, which reduces survival pressures. Under such circumstances, microbial species may rely more on competitive strategies to obtain resources for growth. For instance, bacteria may produce competitive enzymes or secrete compounds to inhibit the growth of rivals [56]. Likewise, fungi may produce antifungal substances to suppress the growth of other species and to secure limited resources [57]. Moreover, differences in nutrient availability associated with elevation may also influence the balance between cooperation and competition within microbial communities. At higher elevations, soils are typically nutrient-poor, requiring microbial taxa to cooperate in order to improve nutrient use efficiency. In contrast, at lower elevations where nutrients are more abundant, microbes may engage in stronger competition to acquire a greater share of the available resources [58].

In summary, elevation influences microbial interactions by altering soil physicochemical properties, nutrient cycling, plant residue composition, and the microenvironment. These changes, in turn, shape the structure and function of microbial communities and modulate the balance between cooperation and competition among species.

### 4.4. Effects of Altitude on Soil Microbial Assembly Mechanisms in Schisandra sphenanthera

To identify the drivers of altitudinal variation in soil microbial communities, we analyzed assembly mechanisms using a null model. Bacterial community assembly was predominantly governed by deterministic processes across all elevations. The contributions of heterogeneous selection and undominated processes declined with increasing altitude, while dispersal limitation increased. This pattern may reflect intensified environmental filtering at higher elevations (e.g., lower temperatures, reduced moisture), which imposes stricter selection pressures, favoring bacterial taxa adapted to harsh conditions [59]. Concurrently, diminished soil nutrient availability at higher altitudes likely exacerbates dispersal limitation by restricting bacterial migration, thereby amplifying the role of local environmental constraints in community assembly [60]. In contrast, fungal community assembly was dominated by stochastic processes. Dispersal limitation decreased with elevation, whereas undominated processes increased. These shifts may arise from heightened environmental unpredictability at higher altitudes, which reduces fungal dispersal capacity and elevates the reliance on stochastic resource availability or random events for survival [61,62]. Correlation analysis further revealed that bacterial βNTI was negatively correlated with soil total phosphorus (STP). This suggests that high phosphorus levels intensify competitive exclusion among bacteria, favoring deterministic dominance by phosphorus-efficient taxa and reducing functional diversity [63]. Conversely, fungal βNTI positively correlated with STP and plant total potassium (PTK), indicating that nutrient-rich conditions enhance fungal functional diversity to optimize organic matter decomposition and nutrient cycling, which are critical for ecosystem stability [64].

This study investigated the effects of different altitudes on the stability of the *Schisandra sphenanthera* soil ecosystem and carbon sequestration, starting from the perspective of soil microbial assembly mechanisms and utilizing microbial co-occurrence networks and pathway analysis. We observed that at lower altitudes, bacterial communities exhibited stronger competitive interactions, while fungal communities showed stronger cooperative interactions. Additionally, microbial communities were positively correlated with soil organic carbon (SOC), whereas altitude was negatively correlated with SOC. However, these correlations themselves do not establish rigorous causation. Therefore, future research should focus on the following three areas: (1) Implementing cross-altitude soil/rhizosphere transplantation experiments to directly assess community adaptability and succession under abrupt environmental changes; (2) Conducting in-depth studies on the dynamics of plant root exudates at different altitudes and their regulatory role in rhizosphere microbial assembly; (3) Integrating metagenomics/meta-transcriptomics with process rate measurements to elucidate the functional implications of community changes. We will also proactively explore key pathways to validate and deepen the findings of this study.

## 5. Conclusions

This study advances our understanding of altitudinal mechanisms shaping soil microbial communities. At lower elevations, bacterial communities exhibited stronger competitive interactions, while fungal communities displayed enhanced cooperation. Conversely, higher elevations fostered cooperative relationships among bacteria but intensified competition among fungi. Furthermore, altitude and soil nutrients showed significant negative correlations with soil carbon content, whereas plant nutrients and fungal diversity positively correlated with soil carbon. Null model analysis revealed that bacterial community assembly was predominantly driven by deterministic processes, whereas fungal assembly followed stochastic processes. These findings provide critical insights into how altitudinal gradients regulate microbial community structure, function, and assembly mechanisms. Future research will delve deeper into the dynamics of plant root exudates at different altitudes and their regulatory role in rhizosphere microbial assembly, providing a scientific basis for the conservation and management of alpine ecosystems.

## Figures and Tables

**Figure 1 biology-14-00944-f001:**
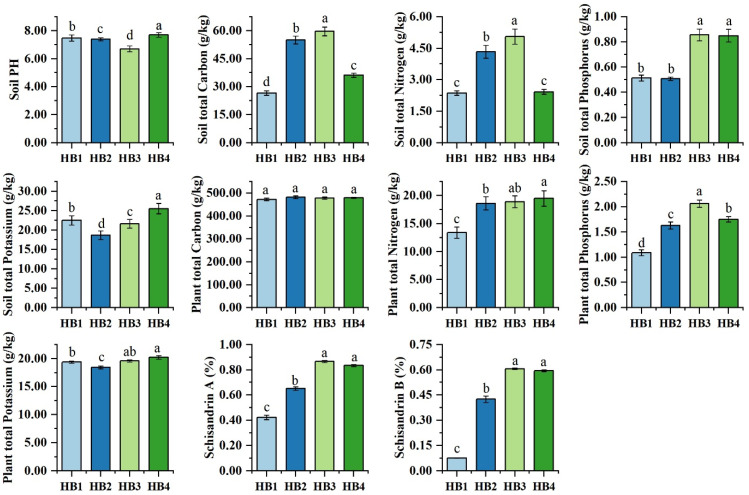
Soil physicochemical properties in the rhizosphere soil of *Schisandra sphenanthera* at different altitude gradients. Note: Different letters indicate significant differences among different altitude gradients (Tukey’s multiple test; *p* < 0.05). HB1—900 m, HB2—1100 m, HB3—1300 m, HB4—1500 m; the same applies to subsequent figures.

**Figure 2 biology-14-00944-f002:**
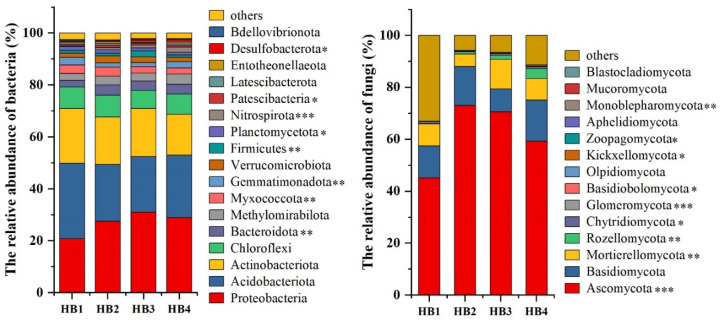
Relative abundance of bacterial and fungal phyla at different altitude gradients. Note: * *p* < 0.05; ** 0.001 < *p* < 0.05; *** *p* < 0.001.

**Figure 3 biology-14-00944-f003:**
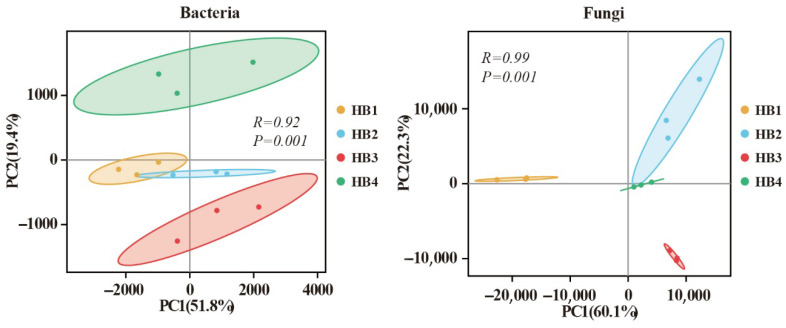
Principal component analysis (PCA) of bacterial and fungal communities at different altitude gradients.

**Figure 4 biology-14-00944-f004:**
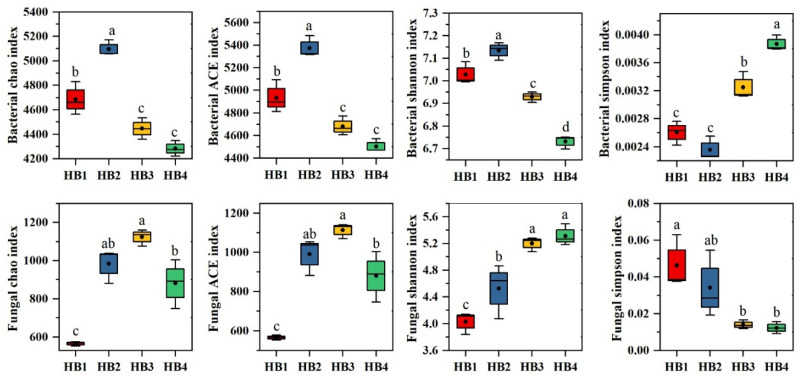
Diversity indices of soil microbial communities. Note: Different letters indicate significant differences between treatments (*p* < 0.05).

**Figure 5 biology-14-00944-f005:**
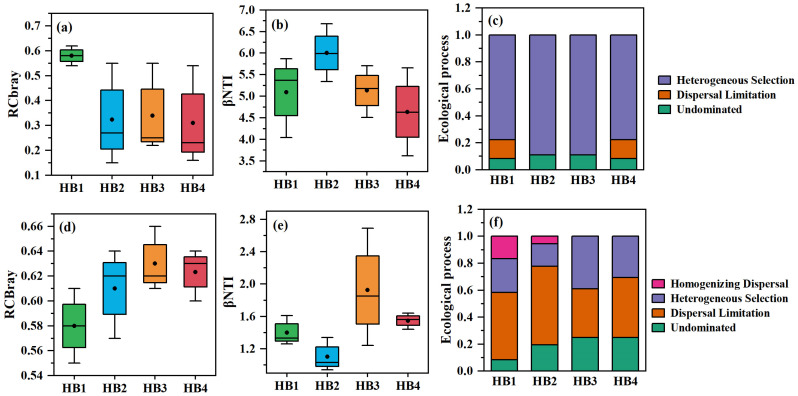
Ecological processes governing soil microbial community assembly. (**a**–**c**) Bacterial community assembly is dominated by deterministic processes (heterogeneous selection), with contributions declining at higher elevations. (**d**–**f**) Fungal community assembly is driven by stochastic processes (dispersal limitation), decreasing with elevation.

**Figure 6 biology-14-00944-f006:**
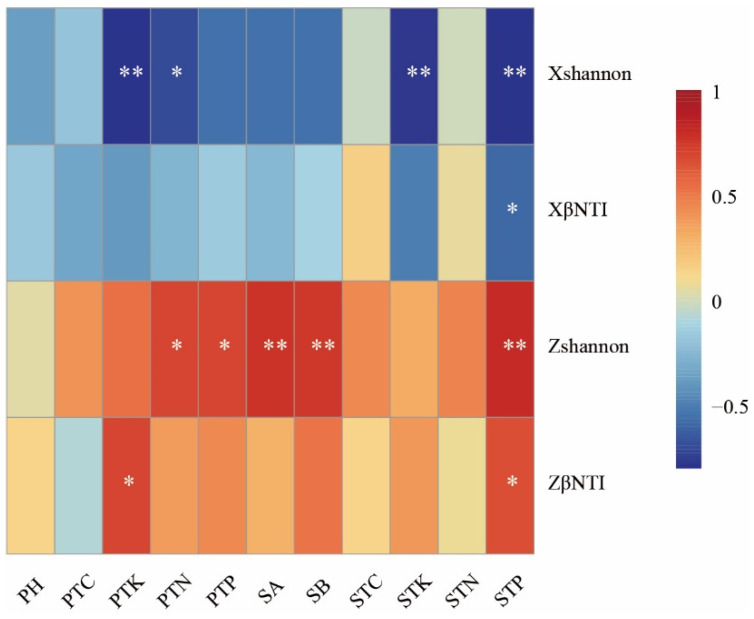
Correlation analysis between nutrient factors and the Beta Nearest Taxon Index (βNTI) of soil microbial communities. Notes: Significance levels: * (*p* < 0.05); ** (0.001 < *p* < 0.05). XShannon: Bacterial Shannon index; ZShannon: Fungal Shannon index. XβNTI: Bacterial assembly mechanism; ZβNTI: Fungal assembly mechanism. STC: Soil total carbon; STK: Soil total potassium; STN: Soil total nitrogen; STP: Soil total phosphorus. PTC: Plant total carbon; PTN: Plant total nitrogen; PTK: Plant total potassium; PTP: Plant total phosphorus. SA: Schisandrin A; SB: Schisandrin B.

**Figure 7 biology-14-00944-f007:**
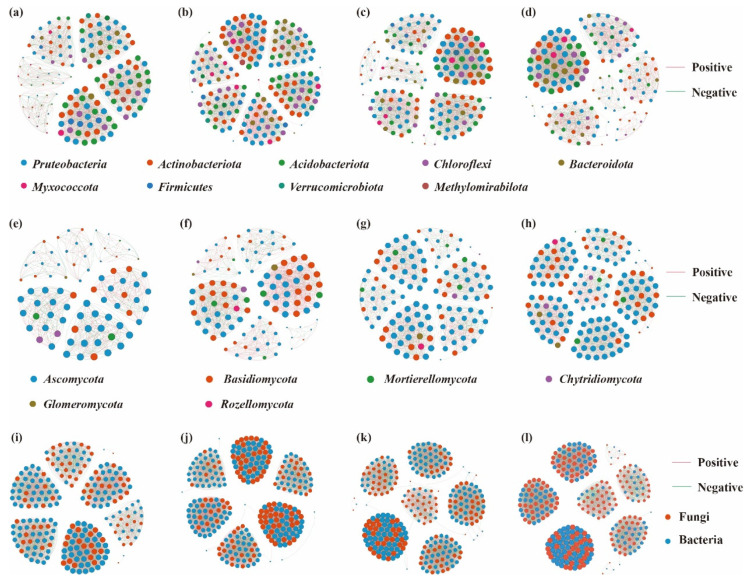
Co-occurrence network analysis of soil microbial communities. Note: Node size represents the degree (number of connected nodes). Red edges indicate positive correlations, while green edges denote negative correlations. Nodes are color-coded based on taxonomic classification (e.g., phylum). (**a**–**d**) represent bacterial networks of HB1–HB4, respectively; (**e**–**h**) represent fungal networks of HB1–HB4; (**i**–**l**) represent bacterial and fungal combined networks of HB1–HB4.

**Figure 8 biology-14-00944-f008:**
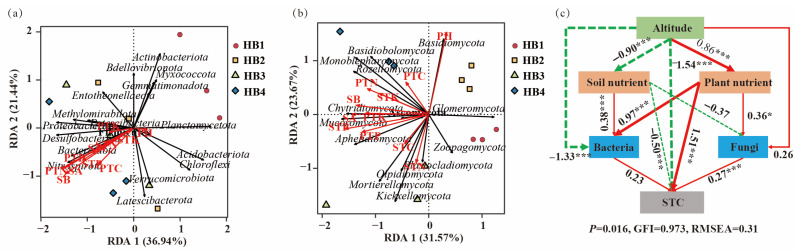
Relationships between soil physicochemical factors and microbial communities. (**a**) RDA analysis of the bacterial community. (**b**) RDA analysis of the fungal community. (**c**) Structural equation model (SEM) analysis. Note: Red solid arrows indicate significant positive correlations, while green dashed arrows indicate negative correlations. The numbers on the arrows represent standardized path coefficients. STC—Soil total carbon; STK—Soil total potassium; STN—Soil total nitrogen; STP—Soil total phosphorus; PTC—Plant total carbon; PTN—Plant total nitrogen; PTK—Plant total potassium; PTP—Plant total phosphorus; SA—Schisandrin A; SB—Schisandrin B. Significance levels are indicated as * *p* < 0.05, *** *p* < 0.001.

**Table 1 biology-14-00944-t001:** Topological parameters of soil microbial co-occurrence networks.

Treatments		Nodes	Links	Positive Edges%	Negative Edges%	Average Degree	Graph Density	Modularity
Bacteria	HB1	150	1978	51.40	48.60	26.37	0.783	0.177
	HB2	185	2591	56.10	43.90	28.01	0.825	0.152
	HB3	164	2363	52.90	47.10	28.82	0.748	0.177
	HB4	147	2017	56.30	43.70	27.44	0.669	0.188
Fungi	HB1	79	527	57.70	42.30	13.34	0.78	0.141
	HB2	112	1198	65.40	34.60	21.39	0.722	0.193
	HB3	130	1322	59.30	40.70	20.34	0.808	0.158
	HB4	160	1917	55.30	44.70	23.96	0.825	0.151
Bacteria × Fungi	HB1	233	4445	52.10	47.90	38.16	0.807	0.164
	HB2	297	7067	53.80	46.20	47.59	0.814	0.161
	HB3	298	7001	52.50	47.50	46.99	0.79	0.158
	HB4	307	7659	52.50	47.50	49.9	0.775	0.163

## Data Availability

The sequence data associated with this project have been deposited in the NCBI database under accession number PRJNA1269727.

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
