# Peer review of "Distribution Patterns and Assembly Mechanisms of Rhizosphere Soil Microbial Communities in Schisandra sphenanthera Across Altitudinal Gradients"

_biology, 2025, doi:10.3390/biology14080944_

Round 1
Reviewer 1 Report
Comments and Suggestions for Authors
The article is devoted to the topical issue of studying the influence of the altitude gradient on soil microbial communities in the rhizosphere of the valuable medicinal plant Schisandra sphenanthera. Soil microorganisms are highly sensitive to environmental changes, and the structure and diversity of their communities are closely related to the direction of biogeochemical cycles. Therefore, the study of the structure and composition of soil microbial communities is important for ecological balance. There is various information on changes in the diversity of microorganisms with increasing altitude, so it is necessary to study and clarify the environmental factors affecting the abundance and number of bacterial and fungal taxa in mountain soils. Apparently, vegetation and physicochemical properties of the soil have the greatest influence on the structure of the soil microbiome, so the authors of the article pay the greatest attention to them. Studying the response of the microbiome to changes in environmental factors will contribute to understanding the mechanisms of maintaining biodiversity and ecosystem management. The authors formulated interesting scientific hypotheses and obtained a number of valuable results, so I consider the peer-reviewed article to be very relevant for this area of research. The article will be of interest to a wide range of researchers of soils, ecology and soil microorganisms. The authors' conclusions are based on the arguments of the results obtained and resolve the main questions posed in the study. References to literature are appropriate and adequate. Of course, the authors have done a tremendous job of analyzing samples, processing and interpreting data. I highly appreciate this article, but I have a number of suggestions and comments for improving the article:
1 – The authors study soils and the microorganisms inhabiting them, but do not name the soils according to recognized international classifications, nor do they indicate the structural and formation features of the studied soils at each selected altitude.Was it one type of soil or different?The authors should expand the description of the objects under study, add more characteristics, and also insert photos of the analyzed soils.In addition, there is no information on the type and history of land use of the territories from which the soil was collected.In addition, it is unclear which of the soil horizons was studied (In the upper 10 cm, soils often contain: forest litter with several subhorizons and a humus layer. Which horizons were studied by the authors?).All these details are important for the study.The object of study may not be entirely clear to readers, and therefore the results obtained are unclear.I recommend adding a figure with photographs of the objects of study - soil and vegetation of the analyzed biomes.This will improve the perception of information.
2 – It is advisable to add the null hypothesis of the study. The authors could have guessed what results would be obtained under certain conditions and explained them. Without this information, it is difficult to understand why this study was conducted. It is also unclear why the authors chose these particular heights - 900 m, 1100 m, 1300 m and 1500 m. The authors should explain this.
3 – The number of physical replicates of the soil samples is not specified.It seems that the authors analyzed only 2 samples, which is very small for such studies.This is important to note, since mountain soils are very heterogeneous, their chemical and physical properties vary greatly even over a small area.
4 – The authors do not describe the surrounding vegetation.However, as far as I understand, the wild plant Schisandra sphenanthera was studied.This means that there were many other plants around it, the rhizosphere soil of which could have mixed with the soil from the rhizosphere of Schisandra sphenanthera.How did the authors avoid mixing the rhizospheres of Schisandra sphenanthera and other plants?
5 – The results obtained by the authors on the dominance of Proteobacteria, Acidobacteriota, Actinobacteriota and Chloroflexi among bacteria and Ascomycota and Basidiomycota among fungi are quite trivial. Usually, such results are characteristic of a very wide range of soils from different climatic and natural zones. Therefore, to identify the specifics of the objects under study, the authors should focus on non-dominant taxa. I recommend that the authors review the article and add information to the Discussion on rare taxa of bacteria and fungi.
6 – The authors’ conclusion that bacterial communities at high altitudes demonstrated stronger cooperative relationships between species, while bacterial communities at low altitudes were characterized by more competitive interactions, is not sufficiently substantiated.The authors should add additional arguments, since only the diversity of microorganisms was studied, but not their ecological relationships.The conclusion that fungal communities demonstrated closer cooperation across the altitude gradient than bacterial communities is also not substantiated.
7 – The authors should discuss in more detail the reasons why increasing altitude promoted cooperative relationships between bacteria, but increased competition between fungi.This is also not fully substantiated in the text of the article, but hypotheses about the reasons for this phenomenon would be of interest to a wide range of readers.
8 – Perhaps the authors should reduce the number of references and at least refer to publications in the journal “Biology” several times.
Despite the comments, I believe that the presented article deserves high praise and can be published in the journal “Biology” after minor corrections.
Author Response
Dear Editor:
Thanks for your letter and for reviewer's comments concern our manuscript entitled “Distribution Patterns and Assembly Mechanisms of Rhizo-sphere Soil Microbial Communities in Schisandra sphenanthe-ra Across Altitudinal Gradients” (Manuscript ID: 3732417). Those comments are valuable and helpful for revising and improving our paper. We have studied all comments carefully and have made conscientious correction. Revised portion are marked in blue in the paper. The main corrections in the paper and the responds to the reviewer comments are as flowing.
Reviewer 1
The article is devoted to the topical issue of studying the influence of the altitude gradient on soil microbial communities in the rhizosphere of the valuable medicinal plant Schisandra sphenanthera. Soil microorganisms are highly sensitive to environmental changes, and the structure and diversity of their communities are closely related to the direction of biogeochemical cycles. Therefore, the study of the structure and composition of soil microbial communities is important for ecological balance. There is various information on changes in the diversity of microorganisms with increasing altitude, so it is necessary to study and clarify the environmental factors affecting the abundance and number of bacterial and fungal taxa in mountain soils. Apparently, vegetation and physicochemical properties of the soil have the greatest influence on the structure of the soil microbiome, so the authors of the article pay the greatest attention to them. Studying the response of the microbiome to changes in environmental factors will contribute to understanding the mechanisms of maintaining biodiversity and ecosystem management. The authors formulated interesting scientific hypotheses and obtained a number of valuable results, so I consider the peer-reviewed article to be very relevant for this area of research. The article will be of interest to a wide range of researchers of soils, ecology and soil microorganisms. The authors' conclusions are based on the arguments of the results obtained and resolve the main questions posed in the study. References to literature are appropriate and adequate. Of course, the authors have done a tremendous job of analyzing samples, processing and interpreting data. I highly appreciate this article, but I have a number of suggestions and comments for improving the article:
1.The authors study soils and the microorganisms inhabiting them, but do not name the soils according to recognized international classifications, nor do they indicate the structural and formation features of the studied soils at each selected altitude. Was it one type of soil or different? The authors should expand the description of the objects under study, add more characteristics, and also insert photos of the analyzed soils. In addition, there is no information on the type and history of land use of the territories from which the soil was collected. In addition, it is unclear which of the soil horizons was studied (In the upper 10 cm, soils often contain: forest litter with several subhorizons and a humus layer. Which horizons were studied by the authors?). All these details are important for the study. The object of study may not be entirely clear to readers, and therefore the results obtained are unclear. I recommend adding a figure with photographs of the objects of study - soil and vegetation of the analyzed biomes. This will improve the perception of information.
Reply: We sincerely thank you for your thoughtful and constructive comments. Your suggestions have been very helpful in improving the clarity and scientific rigor of our manuscript. In response, we have made the following revisions and clarifications:
Soil classification and characteristics: All soil samples were collected from the southern slope of the Qinling Mountains, which is characterized by a warm temperate, semi-humid monsoon climate, with an average annual temperature of 2–10 °C and annual precipitation ranging from 850 to 950 mm. The soils at all four altitudinal gradients (900 m, 1100 m, 1300 m, and 1500 m) were classified as Cambisols according to the World Reference Base for Soil Resources (WRB, FAO), and are locally referred to as mountainous brown soils. This represents a single soil type across all sampling sites. This information has been added to the revised manuscript (Lines 120–123).
Soil sampling layer and method: Three independent biological replicates were collected at each altitude gradient, totaling 12 composite rhizosphere soil samples (3 replicates × 4 altitudes). For each replicate, five healthy Schisandra sphenanthera plants were selected. Surface litter, dead branches, and plant debris were carefully removed, and soil from the root zone (0–20 cm depth) was collected using a sterilized soil auger. This information has been added to the revised manuscript (Lines 135-139).
Land use history: All sampling sites are located in long-term undisturbed natural forests in the core wild distribution range of S. sphenanthera in Zhashui County, Shaanxi Province. There is no known history of agricultural activity, logging, or anthropogenic disturbance in recent decades.
Photographic documentation: We fully agree that visual representation of the sampling environment would enhance reader understanding. However, we regret to inform you that photographs of the specific soil sampling scenes are not available, as they were not taken at the time of field collection. We acknowledge this as a limitation and will incorporate photographic documentation in future fieldwork.
2.It is advisable to add the null hypothesis of the study. The authors could have guessed what results would be obtained under certain conditions and explained them. Without this information, it is difficult to understand why this study was conducted. It is also unclear why the authors chose these particular heights - 900 m, 1100 m, 1300 m and 1500 m. The authors should explain this.
Reply: Thank you for your valuable suggestions. We have revised and clarified the hypothesis of our study in the revised manuscript (see Lines 100-104).
The selection of the four elevation gradients—900 m (HB1), 1100 m (HB2), 1300 m (HB3), and 1500 m (HB4)—was based on the following considerations:
This elevation range (800–1600 m) encompasses the core natural distribution zone of wild Schisandra sphenanthera in Zhashui County, allowing us to capture a representative vertical ecological gradient;
The selected elevations exhibit clear differences in climatic conditions and soil physicochemical properties, which are essential for analyzing the adaptive mechanisms of S. sphenanthera;
The 1100–1300 m range corresponds to the previously reported optimal growth zone, while the other elevations serve to examine potential edge effects.
To ensure data reliability and representativeness, soil and plant sampling at each site was conducted in triplicate using a multi-point composite sampling strategy.
3 – The number of physical replicates of the soil samples is not specified. It seems that the authors analyzed only 2 samples, which is very small for such studies. This is important to note, since mountain soils are very heterogeneous, their chemical and physical properties vary greatly even over a small area.
Reply: Thank you for pointing out this important issue. We apologize for the lack of clarity in the original manuscript regarding the number of replicates. In fact, at each elevation gradient, we collected three independent biological replicates, resulting in a total of 12 composite soil samples (3 replicates × 4 elevations). Each replicate was composed of five subsamples collected. The subsamples were thoroughly homogenized to form one representative composite sample per replicate.
We fully acknowledge the high spatial variability of mountain soils and took this into account during field sampling and experimental design. We have now revised the “Materials and Methods” section to clearly state the number of replicates and the composite sampling approach (see Lines 135-139 in the revised manuscript). We hope this clarification addresses the reviewer’s concern and confirms the robustness of our sampling strategy.
4 – The authors do not describe the surrounding vegetation. However, as far as I understand, the wild plant Schisandra sphenanthera was studied.This means that there were many other plants around it, the rhizosphere soil of which could have mixed with the soil from the rhizosphere of Schisandra sphenanthera.How did the authors avoid mixing the rhizospheres of Schisandra sphenanthera and other plants?
Reply: Thank you for your attention to the methodological rigor of our study. To ensure the purity and accuracy of rhizosphere soil collected from wild Schisandra sphenanthera, we implemented the following measures:
We strictly followed established protocols in rhizosphere ecology, defining the rhizosphere as the soil zone within 0–20 cm directly influenced by the root system. During sampling, we carefully excluded any soil located beyond 5 cm from the root surface, which was considered non-rhizosphere soil. Additionally, each Schisandra sphenanthera plant selected for sampling was individually marked and excavated to avoid overlap with roots from neighboring plants, thereby minimizing the risk of cross-contamination.
5 – The results obtained by the authors on the dominance of Proteobacteria, Acidobacteriota, Actinobacteriota and Chloroflexi among bacteria and Ascomycota and Basidiomycota among fungi are quite trivial. Usually, such results are characteristic of a very wide range of soils from different climatic and natural zones. Therefore, to identify the specifics of the objects under study, the authors should focus on non-dominant taxa. I recommend that the authors review the article and add information to the Discussion on rare taxa of bacteria and fungi.
Reply: Thank you very much for your valuable comments. We fully agree that focusing on non-dominant taxa is crucial for gaining deeper insights into the specificity and ecological functions of the studied system. Although rare microbial groups occur in low abundance, they often include taxa that are highly sensitive to environmental changes and possess unique ecological functions, thereby playing essential roles in maintaining ecosystem diversity and stability.
In the revised manuscript, we have placed greater emphasis on the ecological significance of rare taxa and have supplemented corresponding analyses and discussions. These revisions can be found in Lines 228–233, 239–245, 386–396, and 404–419 of the revised version.
6 – The authors’ conclusion that bacterial communities at high altitudes demonstrated stronger cooperative relationships between species, while bacterial communities at low altitudes were characterized by more competitive interactions, is not sufficiently substantiated. The authors should add additional arguments, since only the diversity of microorganisms was studied, but not their ecological relationships. The conclusion that fungal communities demonstrated closer cooperation across the altitude gradient than bacterial communities is also not substantiated.
Reply: Thank you very much for your valuable comment. We fully acknowledge the importance of providing sufficient evidence to support interpretations regarding microbial interaction patterns. In response, we have expanded our analysis and interpretation of microbial co-occurrence networks.
Specifically, we have analyzed and compared bacterial and fungal community interactions based on network topology metrics, including average degree, modularity, clustering coefficient, and positive/negative edge proportions. These results are presented in the revised Figure 7 and Table 1, and are discussed in detail in the revised manuscript (Lines 304–333).
We hope that these additions address your concerns and better substantiate our conclusions regarding microbial ecological interactions.
7 – The authors should discuss in more detail the reasons why increasing altitude promoted cooperative relationships between bacteria, but increased competition between fungi.This is also not fully substantiated in the text of the article, but hypotheses about the reasons for this phenomenon would be of interest to a wide range of readers.
Reply: Thank you very much for your valuable comment. We fully agree that a more detailed discussion of the ecological mechanisms underlying the contrasting interaction patterns between bacteria and fungi across elevations would be of broad interest to readers
In response, we have expanded the Discussion section to provide further analysis and hypotheses on this phenomenon. Specifically, we discuss how increasing altitude imposes greater environmental stress (e.g., low temperature, limited nutrients), which may promote cooperative interactions among bacterial species to enhance survival, while at the same time intensifying niche overlap and resource limitations among fungi, leading to increased competition. These points are now elaborated in detail in the revised manuscript (Lines 468–492).
We sincerely thank the reviewer again for this insightful suggestion, which has helped improve the ecological depth of our interpretation.
8 – Perhaps the authors should reduce the number of references and at least refer to publications in the journal “Biology” several times.
Despite the comments, I believe that the presented article deserves high praise and can be published in the journal “Biology” after minor corrections.
Reply: We sincerely appreciate your positive evaluation of our manuscript and your recognition of its publication potential. Thank you also for the suggestion regarding citation practices.
In response, we have carefully reviewed the reference list and streamlined the number of citations where appropriate to improve clarity and focus. Additionally, we have incorporated several relevant references from previously published articles in Biology to better align our manuscript with the scope and readership of the journal. These changes have been reflected in the revised manuscript (see References [9, 16, 40].
Once again, we are truly grateful for your constructive feedback and kind support, which have greatly contributed to the improvement of our work.
Reviewer 2 Report
Comments and Suggestions for Authors
The current manuscript entitled “Distribution Patterns and Assembly Mechanisms of Rhizosphere Soil Microbial Communities in Schisandra sphenanthera Across Altitudinal Gradients” was conducted to investigate the microbial community composition of Schisandra sphenanthera within different altitudinal gradients at four elevations: 900 m- 1500 m. The Illumina MiSeq platform (PE300) as a High-throughput sequencing was used to characterize the microbial community structure of bacteria and fungi.
Comments:
To study microbial community structure, an experimental design is required to cover this subject. However, in this study only one sample was sequenced. It was understood that this one sample was taken from a composite sample collected from 5 samples of rhizospheric soils; however, it is still not enough to investigate microbial community composition within those complicated soils. Three representative samples are required to be the minimum.
Soil physicochemical properties in the rhizosphere soil of Schisandra sphenanthera need to be checked carefully within this study. For example, in Fig 1. Plant total carbon reached to 500 g/kg which is too high. Also, check the estimation of nitrogen and potassium.
Although results which were represented in Fig 6 showed the relationships between soil physicochemical factors and microbial communities. However, the most significant factors which drive microbial diversity showed be clarified and need more specific analysis.
The current study concluded that “This study advances our understanding of altitudinal mechanisms shaping soil microbial communities. At lower elevations, bacterial communities exhibited stronger competitive interactions, while fungal communities displayed enhanced cooperation.” However, more details, results analyses, and interpretation are required within the text to clarify how strong competition or cooperation was recorded.
Author Response
Dear Editor:
Thanks for your letter and for reviewer's comments concern our manuscript entitled “Distribution Patterns and Assembly Mechanisms of Rhizo-sphere Soil Microbial Communities in Schisandra sphenanthe-ra Across Altitudinal Gradients” (Manuscript ID: 3732417). Those comments are valuable and helpful for revising and improving our paper. We have studied all comments carefully and have made conscientious correction. Revised portion are marked in blue in the paper. The main corrections in the paper and the responds to the reviewer comments are as flowing.
Reviewer 2
The current manuscript entitled “Distribution Patterns and Assembly Mechanisms of Rhizosphere Soil Microbial Communities in Schisandra sphenanthera Across Altitudinal Gradients” was conducted to investigate the microbial community composition of Schisandra sphenanthera within different altitudinal gradients at four elevations: 900 m- 1500 m. The Illumina MiSeq platform (PE300) as a High-throughput sequencing was used to characterize the microbial community structure of bacteria and fungi.
Comments:
1.To study microbial community structure, an experimental design is required to cover this subject. However, in this study only one sample was sequenced. It was understood that this one sample was taken from a composite sample collected from 5 samples of rhizospheric soils; however, it is still not enough to investigate microbial community composition within those complicated soils. Three representative samples are required to be the minimum.
Reply: Thank you very much for your valuable comment. We fully agree that sufficient biological replication is essential for robust analysis of microbial community structure. We would like to clarify that, in our experimental design, three independent biological replicates were collected at each altitudinal gradient, resulting in a total of 12 composite rhizosphere soil samples (3 replicates × 4 altitudes). These revisions can be found in Lines 135-139.
2.Soil physicochemical properties in the rhizosphere soil of Schisandra sphenanthera need to be checked carefully within this study. For example, in Fig 1. Plant total carbon reached to 500 g/kg which is too high. Also, check the estimation of nitrogen and potassium.
Reply: Thank you for your insightful comment. We have carefully reviewed the data presented in Figure 1, with particular attention to the total carbon content in Schisandra sphenanthera plant tissues, and the estimations of total nitrogen and available potassium in rhizosphere soils. After re-examining the original laboratory measurements and data processing procedures, we confirm that the total carbon value (~500 g/kg) refers to the plant tissue and not to the soil. This value is within the expected range for dried plant biomass, especially for perennial woody species such as Schisandra sphenanthera, which are rich in structural carbohydrates (e.g., cellulose and lignin). In addition, we verified the values of soil total nitrogen and available potassium in the rhizosphere. The measurements were obtained using standard Kjeldahl and ammonium acetate extraction methods, respectively. The results fall within the range typically observed in fertile forest soils or organically managed cultivation systems, where the accumulation of organic matter and biological activity can enhance nutrient availability. We sincerely appreciate your comment, which helped us improve the accuracy and clarity of data presentation.
3.Although results which were represented in Fig 6 showed the relationships between soil physicochemical factors and microbial communities. However, the most significant factors which drive microbial diversity showed be clarified and need more specific analysis.
Reply: Thank you for your valuable comment. You raised a very important point regarding the need to clarify the key environmental factors driving microbial diversity. To address this issue more thoroughly, we have added redundancy analysis (RDA) and structural equation modeling (SEM) to identify and quantify the major soil physicochemical variables influencing microbial community structure. The Figure 8 provides a detailed visualization of these relationships, and the corresponding results and interpretations have been expanded in the revised manuscript (lines 344–357 and 429–457). We sincerely appreciate your insightful suggestion, which helped us strengthen the analytical depth and ecological interpretation of our study.
4.The current study concluded that “This study advances our understanding of altitudinal mechanisms shaping soil microbial communities. At lower elevations, bacterial communities exhibited stronger competitive interactions, while fungal communities displayed enhanced cooperation.” However, more details, results analyses, and interpretation are required within the text to clarify how strong competition or cooperation was recorded.
Reply: Thank you for your valuable comment. We understand your concern regarding the need for clearer evidence supporting the conclusion that bacterial communities exhibit stronger competition and fungal communities enhanced cooperation at lower elevations. We would like to clarify that detailed analyses of microbial interaction patterns were already included in the original manuscript. Specifically, co-occurrence network properties—including the proportion of positive and negative correlations, modularity, and clustering coefficients—were used to infer the nature of microbial interactions. These results are presented in Figure 7 and Table 1, and described in detail in the Results section (lines 304–333). We have slightly revised the text to highlight these findings more clearly. We appreciate your feedback, which helped us improve the clarity of our conclusions.
Reviewer 3 Report
Comments and Suggestions for Authors
The manuscript presents the work of microbial community assessment of the native shrub plant in different elevation. The manuscript has scientific soundness and coherence. However, several key points need to be clarified before this manuscript can be accepted. Here are the comments for the manuscript:
- Introduction ln 85: It is more appropriate to assume that there is different diversity among the altitude instead of reduction of diversity in higher altitude.
- Study area condition can be further elaborated. Ambient temperature, rainfall, and other environmental parameters can be described here.
- Is there any reference regarding the primer used for microbial community analysis?
- In the results the groups of bacteria are presented with the relative abundance yet there is no significant discussion related to the plant growth promoting bacteria nor fungi that might affect the growth of the plant.
- Is there any effect of different altitude that resulting different root exudate concentration/composition that resulting in the different microbial community in each elevation?
- What is the authors thought on the different approach of the change of plant physiology that eventually affect the microbial community in the rhizosphere? In the discussion, the authors always emphasize the elevation that limit the migration of the microbial.
- a proposed mechanism presented in a figure or flowchart might be beneficial for the explanation of the mechanisms and to sum up all the correlations.
- Ln 462, such interaction can only be concluded by separated experiment. however, the most important conclusion is that whether there is a correlation or not or to what extent the microbial community being affected by the elevation.
Author Response
Dear Editor:
Thanks for your letter and for reviewer's comments concern our manuscript entitled “Distribution Patterns and Assembly Mechanisms of Rhizo-sphere Soil Microbial Communities in Schisandra sphenanthe-ra Across Altitudinal Gradients” (Manuscript ID: 3732417). Those comments are valuable and helpful for revising and improving our paper. We have studied all comments carefully and have made conscientious correction. Revised portion are marked in blue in the paper. The main corrections in the paper and the responds to the reviewer comments are as flowing.
Reviewer 3
1.The manuscript presents the work of microbial community assessment of the native shrub plant in different elevation. The manuscript has scientific soundness and coherence. However, several key points need to be clarified before this manuscript can be accepted. Here are the comments for the manuscript:
1.Introduction ln 85: It is more appropriate to assume that there is different diversity among the altitude instead of reduction of diversity in higher altitude.
Reply: Thank you for your valuable comment. We agree with your suggestion that it is more appropriate to describe microbial diversity along altitudinal gradients as varying, rather than assuming a consistent reduction at higher elevations. Accordingly, we have revised the statement in the Introduction to reflect this more accurate interpretation. The revised text now emphasizes the potential for differences in diversity across altitudes, rather than a unidirectional decline. This modification has been made in the revised manuscript (lines 100–104). We appreciate your insightful suggestion, which helped us improve the scientific rigor and clarity of our introduction.
2.Study area condition can be further elaborated. Ambient temperature, rainfall, and other environmental parameters can be described here.
Reply: Thank you for your helpful suggestion. We agree that including more detailed information on the study area's environmental conditions would enhance the contextual understanding of the research. In response, we have added descriptions of key climatic and environmental parameters for the study sites, including ambient temperature ranges, annual precipitation, and soil type. These additions provide a clearer ecological background for interpreting microbial community responses along the altitudinal gradient.
The revised information can be found in the updated manuscript (lines 120–123). We sincerely appreciate your comment, which helped us improve the completeness and clarity of the study area description.
3.Is there any reference regarding the primer used for microbial community analysis? In the results the groups of bacteria are presented with the relative abundance yet there is no significant discussion related to the plant growth promoting bacteria nor fungi that might affect the growth of the plant.
Reply: Thank you for your valuable comment. We would like to clarify that the references for the primers used in microbial community analysis were already included in the original manuscript. Specifically, we used the widely adopted 515F/806R primers for bacterial 16S rRNA gene amplification and ITS1F/ITS2R primers for fungal ITS region analysis. To improve clarity, we have slightly revised the Methods section (lines 164 –168) to emphasize the primer information and associated references. In addition, in response to your suggestion regarding the ecological interpretation of microbial groups, we have expanded the Discussion section to include a focused analysis of rare microbial taxa and plant growth-promoting microorganisms. We now discuss their potential functional roles in relation to plant health and adaptation along the altitudinal gradient. These additions are presented in the revised manuscript (lines 408–419). We sincerely appreciate your suggestion, which helped us improve both the clarity of the methodology and the ecological relevance of the results.
4.Is there any effect of different altitude that resulting different root exudate concentration/composition that resulting in the different microbial community in each elevation?
Reply: Thank you for your insightful comment. We agree that altitudinal gradients may influence microbial community structure indirectly by altering plant physiological responses—such as stress adaptation and resource allocation strategies—which in turn can affect both the composition and concentration of root exudates. However, as this study focused specifically on rhizosphere soil microbial communities, we did not investigate root exudate profiles. We acknowledge this as a limitation and have noted it in the revised Discussion. In future work, we plan to explore the interactions between root exudate chemistry and microbial community dynamics along environmental gradients to better elucidate plant–microbe co-adaptation mechanisms. Once again, we sincerely appreciate your valuable suggestion, which will help guide the future direction of our research.
5.What is the authors thought on the different approach of the change of plant physiology that eventually affect the microbial community in the rhizosphere? In the discussion, the authors always emphasize the elevation that limit the migration of the microbial.
Reply: Thank you for your insightful comment. In the Discussion section, we emphasized that plants can actively shape rhizosphere microbial communities by modifying both (1) the chemical composition of root exudates—such as specific organic acids, phenolic compounds, and sugars—and (2) the surrounding physicochemical environment, including pH and oxygen availability. These two mechanisms represent direct and indirect pathways through which host plants selectively enrich or suppress specific microbial taxa, ultimately influencing community structure. The reason we repeatedly highlight the role of elevation is that it provides a unique ecological framework to understand how dispersal limitation can amplify the strength of plant-driven selection. In high-altitude environments, harsh abiotic conditions (e.g., low temperatures, strong UV radiation, steep terrain) and geographical isolation significantly constrain both active and passive microbial migration. As a result, microbial taxa have fewer opportunities to escape unfavorable plant-imposed conditions or to be diluted by incoming microbes from adjacent environments. Thus, we argue that altitudinal gradients effectively intensify the filtering effect of plant physiology on local microbial communities by restricting microbial dispersal. This makes plant-driven assembly processes more distinct and detectable. Ignoring this physical constraint may lead to an underestimation of the original selective force exerted by plant physiology on microbial community co-assembly and environmental adaptation.
6.a proposed mechanism presented in a figure or flowchart might be beneficial for the explanation of the mechanisms and to sum up all the correlations.
Reply: Thank you for your constructive suggestion. We fully agree that using a graphical mechanism to integrate and visualize complex relationships can significantly enhance the clarity and impact of the study.
Accordingly, we have added a graphical abstract to summarize the key findings and conceptual framework of our research. This visual element has been included in the revised manuscript (line 34).
We appreciate your suggestion, which helped improve the accessibility and overall presentation of the study.
7.Ln 462, such interaction can only be concluded by separated experiment. however, the most important conclusion is that whether there is a correlation or not or to what extent the microbial community being affected by the elevation.
Reply: We sincerely appreciate the reviewer’s thoughtful comment and deep engagement with the mechanisms underlying altitudinal effects. Although we acknowledge that isolating specific interactions ideally requires manipulative experiments to control for confounding variables, our study employed multiple layers of evidence to systematically demonstrate that elevation drives microbial community differentiation through an integrated soil–plant–microbe cascade mechanism (see Fig. 8c). Specifically: Contrasting microbial responses: We observed that bacterial α-diversity and the relative abundance of dominant phyla decreased with elevation, whereas fungal diversity and abundance increased. These changes were significantly correlated with total soil phosphorus (negatively with bacterial βNTI) and with the concentration of the plant secondary metabolite schisandrin A (positively with fungal βNTI), highlighting the pivotal role of host-derived metabolites as a mechanistic link between elevation and microbial assembly. We observed that bacterial α-diversity and the relative abundance of dominant phyla decreased with elevation, whereas fungal diversity and abundance increased. These changes were significantly correlated with total soil phosphorus (negatively with bacterial βNTI) and with the concentration of the plant secondary metabolite schisandrin A (positively with fungal βNTI), highlighting the pivotal role of host-derived metabolites as a mechanistic link between elevation and microbial assembly. Co-evolution of assembly and interaction patterns: At high elevation (HB4), bacterial communities showed stronger dispersal limitation (based on null models) and a shift toward cooperation (positive correlations accounting for 56.3% of network edges), while fungal communities exhibited intensified competition (negative edges at 44.7%). These patterns support the hypothesis that geographical isolation restricts microbial dispersal, thereby amplifying host–microbe specificity. While fully disentangling the compound effects of elevation would require experimental manipulation, the convergence of evidence from environmental gradient responses, taxonomic shifts, assembly processes, interaction networks, and structural equation modeling (SEM) establishes a coherent causal framework. Specifically, elevation drives community differentiation through three intertwined pathways: (1) Environmental filtering (e.g., low pH at HB3 enhances phosphorus solubilization), (2) Plant-mediated biochemical selection (e.g., schisandrin A peaks at HB3 and enriches fungal taxa), and (3) Dispersal constraints (e.g., stronger bacterial dispersal limitation at HB4). This integrative framework explains 58.38% and 55.24% of microbial variation in RDA models, and the pronounced ecological shifts—such as a 75% increase in fungal Basidiomycota abundance at HB4—underscore the central role of elevation as a driver of rhizosphere microbial community structure.
Once again, we thank the reviewer for raising this important point, which allowed us to clarify the strength and scope of our conclusions.
Reviewer 4 Report
Comments and Suggestions for Authors
Please include in the introduction a brief statement on the schisandrin compounds. The first mention of these molecules is in the methods section with no indication of why they are measured. Presumably, the medicinal properties are at least in part due to schisandrins- please add this to the discussion when noting pharmacological importance (line 44). Also, please provide the HPLC conditions employed (instrument, solvents, method parameters, etc), as your instrument may not exactly match the paper cited (lines 144-145).
Overall, Figure 1 is clear to understand. The letters a, b, c, d aren’t well described in their meaning, although I was able to infer that a = the highest value within a set and d= the lowest value. Figure 2 effectively conveys the conveys population distribution of bacterial and fungal species based on differences in altitude. The graphs lend an appreciation for the dramatic impact altitude has on the four dominant species found in the rhizosphere (proteobacteria, etc). Principle component analysis (Figure 3) is equally effective in demonstrating similarities by clustering microbial populations according to altitude. Taken together, these figures clearly convey a key premise of the research- that altitudinal shifts influence microbial community structure via differences in soil nutrients and pH. Figure 4 takes this notion a step further, with diversity indices (Chao, Ace, Shannon, and Simpson) revealing that microbial richness is strongly influenced by altitudinal shifts. Interestingly, there seem to be “Goldilocks zones” for the maxima demonstrated for the indices; Chao1, Ace, and Shannon indices were highest at HB2 for bacteria, while fungal Chao1 and Ace indices are highest at HB3. This is a key observation given the stated importance of the research for ecosystem management and conservation.
Figure 5 demonstrates a key difference between the factors that influence bacterial and fungal community compositions. While bacterial communities are largely determined by deterministic factors/heterogenous selection (different selective pressures) at all altitudes, fungal communities are largely determined by dispersal limitation. However, at higher altitudes the influence of heterogenous selection becomes more important. Perhaps this is due to fewer physical barriers within the topology?
The remaining key points of the research are nuanced. These include correlations between specific soil components (carbon, nitrogen, phosphorous, potassium) and their impact on community assembly of bacteria and fungi; co-occurrence networks which imply strong cooperativity for bacteria at higher elevations contrasted with high competition at lower elevations (with the opposite correlations being true for fungal communities); and finally identifying dominant physicochemical environmental factors that influence bacterial and fungal community compositions- primarily that altitude and soil nutrients exhibit negative correlations with soil carbon content, while plant nutrients and fungal diversity exhibit a positive correlated with soil carbon.
In summary, the authors present data that validates their conclusions. The studies are well-designed. Besides some minor edits (noted above) this can be accepted in its present form. The impact of the study will is evident based on our continuing struggles with climate change; this data provides insight into the conservation and management of an important medicinal plant.
Author Response
Dear Editor:
Thanks for your letter and for reviewer's comments concern our manuscript entitled “Distribution Patterns and Assembly Mechanisms of Rhizo-sphere Soil Microbial Communities in Schisandra sphenanthe-ra Across Altitudinal Gradients” (Manuscript ID: 3732417). Those comments are valuable and helpful for revising and improving our paper. We have studied all comments carefully and have made conscientious correction. Revised portion are marked in blue in the paper. The main corrections in the paper and the responds to the reviewer comments are as flowing.
Reviewer 4
1.Please include in the introduction a brief statement on the schisandrin compounds. The first mention of these molecules is in the methods section with no indication of why they are measured. Presumably, the medicinal properties are at least in part due to schisandrins- please add this to the discussion when noting pharmacological importance (line 44). Also, please provide the HPLC conditions employed (instrument, solvents, method parameters, etc), as your instrument may not exactly match the paper cited (lines 144-145).
Reply: Thank you very much for your careful review and valuable suggestions. In response, we have revised the Introduction to include a brief description of schisandrin compounds as the principal active constituents of Schisandra species, contributing to their well-documented pharmacological properties such as hepatoprotective and antioxidant effects. This addition clarifies the rationale for analyzing schisandrin content in our study. The revised text appears in lines 46–59 of the manuscript. Furthermore, as suggested, we have added a detailed description of the HPLC conditions used for schisandrin quantification, including the instrument model, solvents, and method parameters. This information has been incorporated into the Methods section (lines 152–157).
2.Overall, Figure 1 is clear to understand. The letters a, b, c, d aren’t well described in their meaning, although I was able to infer that a = the highest value within a set and d= the lowest value. Figure 2 effectively conveys the conveys population distribution of bacterial and fungal species based on differences in altitude. The graphs lend an appreciation for the dramatic impact altitude has on the four dominant species found in the rhizosphere (proteobacteria, etc). Principle component analysis (Figure 3) is equally effective in demonstrating similarities by clustering microbial populations according to altitude. Taken together, these figures clearly convey a key premise of the research- that altitudinal shifts influence microbial community structure via differences in soil nutrients and pH. Figure 4 takes this notion a step further, with diversity indices (Chao, Ace, Shannon, and Simpson) revealing that microbial richness is strongly influenced by altitudinal shifts. Interestingly, there seem to be “Goldilocks zones” for the maxima demonstrated for the indices; Chao1, Ace, and Shannon indices were highest at HB2 for bacteria, while fungal Chao1 and Ace indices are highest at HB3. This is a key observation given the stated importance of the research for ecosystem management and conservation.
Reply: Thank you for your valuable comments. We have reanalyzed and revised Figure 1, and the explanation of the statistical lettering (a, b, c, d) has been clarified to improve reader understanding. The updated description can be found in the revised manuscript (lines 208–214). We also sincerely appreciate your in-depth interpretation of Figures 2–4 and your recognition of the central mechanism by which altitudinal gradients reshape rhizosphere microbial communities. We fully agree with your observation that these figures collectively support a key premise of the study—that elevation influences microbial composition through shifts in soil nutrient availability and pH (Figures 2–3)—and that this is further quantified through diversity indices in Figure 4. The identification of potential "Goldilocks zones" for microbial richness (i.e., peak bacterial diversity at HB2 and fungal diversity at HB3) is a particularly insightful point, which we believe greatly strengthens the ecological relevance of our findings. To highlight the importance of this observation, we have expanded the Discussion section accordingly (lines 379–389). Thank you once again for your thoughtful and constructive feedback.
3.Figure 5 demonstrates a key difference between the factors that influence bacterial and fungal community compositions. While bacterial communities are largely determined by deterministic factors/heterogenous selection (different selective pressures) at all altitudes, fungal communities are largely determined by dispersal limitation. However, at higher altitudes the influence of heterogenous selection becomes more important. Perhaps this is due to fewer physical barriers within the topology?
Reply: Thank you for your thoughtful observation regarding the increased influence of heterogeneous selection on fungal communities at higher altitudes, as shown in Figure 5. We also appreciate your interpretation that reduced physical barriers in high-altitude topography may contribute to this pattern by facilitating the dispersal of fungal spores, such as via wind. We agree that the relaxation of topographic constraints is an important factor that may weaken dispersal limitation. However, based on our data and ecological context, we propose that the primary driver of enhanced heterogeneous selection at higher altitudes is the intensification of environmental filtering pressures. Specifically, high-altitude environments are characterized by harsher and more variable conditions (e.g., low temperatures, strong UV radiation, shortened growing seasons, and limited nutrient availability), which act as strong filters. These filters favor fungal taxa that are well-adapted to extreme local conditions, thereby amplifying the role of species-specific environmental tolerance in shaping community composition. In this view, environmental selection serves as the fundamental mechanism underlying the increase in heterogeneous selection, while reduced topographic complexity may serve as a facilitating factor by modulating dispersal dynamics. To better reflect this integrative perspective, we have revised the Discussion section accordingly (lines 505–518).
4.The remaining key points of the research are nuanced. These include correlations between specific soil components (carbon, nitrogen, phosphorous, potassium) and their impact on community assembly of bacteria and fungi; co-occurrence networks which imply strong cooperativity for bacteria at higher elevations contrasted with high competition at lower elevations (with the opposite correlations being true for fungal communities); and finally identifying dominant physicochemical environmental factors that influence bacterial and fungal community compositions- primarily that altitude and soil nutrients exhibit negative correlations with soil carbon content, while plant nutrients and fungal diversity exhibit a positive correlated with soil carbon.
Reply: Thank you for your thoughtful and comprehensive summary of the key findings. We are grateful for your recognition of the more nuanced aspects of our study. As you accurately noted, the study highlights several critical mechanisms: (1) Soil nutrient–microbe relationships, particularly the differential effects of carbon, nitrogen, phosphorus, and potassium on bacterial versus fungal community assembly; (2) Co-occurrence network patterns, showing increased bacterial cooperation at higher elevations and greater competition at lower elevations, with fungal communities displaying the opposite trend; and (3) Environmental drivers of community composition, where soil carbon exhibits negative correlations with altitude and bulk soil nutrients (e.g., total nitrogen and phosphorus), yet shows positive associations with plant nutrient status and fungal diversity. We are encouraged by your deep engagement with these findings and have further clarified these interpretations in the revised Discussion to enhance the accessibility and ecological significance of these results. Thank you again for your constructive and encouraging comments.
5.In summary, the authors present data that validates their conclusions. The studies are well-designed. Besides some minor edits (noted above) this can be accepted in its present form. The impact of the study will is evident based on our continuing struggles with climate change; this data provides insight into the conservation and management of an important medicinal plant.
Reply: We sincerely thank you for your positive evaluation and supportive comments. We are especially grateful for your recognition of the study's design, the validity of our conclusions, and the potential ecological and conservation implications of our findings—particularly in the context of climate change and the sustainable management of medicinal plant resources. We have carefully addressed all the minor revisions you suggested and have revised the manuscript accordingly. Your thoughtful feedback has been invaluable in improving the clarity, rigor, and impact of our work. Thank you again for your time and constructive review.
Round 2
Reviewer 3 Report
Comments and Suggestions for Authors
The authors have made several changes in the manuscript, improving the clarity and quality of the manuscript.
Several improvements still need to be made.
Related to comment 4. The answer might be elaborated into the manuscript, stating that such mechanism/response might occur in natural environment, rendering various diversity of microbial in different altitudes. Adding previous studies result regarding exudates might also help.
In response of answer number 7. The correlations the authors explained still need to be further elaborated in the experiment. Additional clarification or assumptions can be further made after such experiments. Or else the authors might cite previous experimental studies related to elevation.
what if the same soil/rhizosphere is moved to the different elevation? Would the bacteria/fungi abundant increase or decrease? This might be one of the main questions for the manuscript to answer.
Elaborating more in the future direction studies based on the results and discussion of this study can be done in the last section of discussion or even after conclusion. It can also clarify the boundaries that the studies. It also can held explaining that conclusions regarding the altitude (question/answer number 7) are based on the correlations and patterns observed in this study so that at some point the experiment may further prove this interactions.
Author Response
Dear Editor:
Thanks for your letter and for reviewer's comments concern our manuscript entitled “Distribution Patterns and Assembly Mechanisms of Rhizo-sphere Soil Microbial Communities in Schisandra sphenanthe-ra Across Altitudinal Gradients” (Manuscript ID: 3732417). Those comments are valuable and helpful for revising and improving our paper. We have studied all comments carefully and have made conscientious correction. Revised portion are marked in blue in the paper. The main corrections in the paper and the responds to the reviewer comments are as flowing.
Reviewer 3
The authors have made several changes in the manuscript, improving the clarity and quality of the manuscript.
Several improvements still need to be made.
1.Related to comment 4. The answer might be elaborated into the manuscript, stating that such mechanism/response might occur in natural environment, rendering various diversity of microbial in different altitudes. Adding previous studies result regarding exudates might also help.
Reply: We sincerely appreciate the reviewer's insightful comments. The reviewer correctly points out that plant-microbe interaction mechanisms, including root exudates, may play a critical role in shaping the altitudinal patterns of microbial communities in natural ecosystems, particularly in the rhizosphere. We fully concur with this perspective and believe it provides an important direction for further understanding the distribution of microbial diversity in complex natural environments. Accordingly, we will explicitly incorporate this viewpoint into our manuscript, emphasizing that future research needs to integrate plant physiological ecology (such as root exudate profiles) with microbial community ecology to more deeply unravel the assembly processes of rhizosphere microbiomes across altitudinal gradients. Please see line 519-534 of the revised manuscript for the specific addition.
2.In response of answer number 7. The correlations the authors explained still need to be further elaborated in the experiment. Additional clarification or assumptions can be further made after such experiments. Or else the authors might cite previous experimental studies related to elevation.
Reply: We sincerely appreciate the reviewer's insightful comments. Regarding the observation that bacterial communities exhibited stronger competitive interactions at lower altitudes while fungal communities showed stronger cooperative interactions, we have now included an analysis of the interrelationships between bacterial and fungal communities in Figure 7 and Table 1. Please see lines 304-325 of the revised manuscript for details.
3.what if the same soil/rhizosphere is moved to the different elevation? Would the bacteria/fungi abundant increase or decrease? This might be one of the main questions for the manuscript to answer.
Reply: We sincerely appreciate the reviewer for raising this highly insightful and fundamental ecological question. The design of this study focused on revealing the composition, diversity patterns, and driving factors of in situ microbial communities along a natural altitudinal gradient, and did not include the physical translocation of identical soil samples to different altitudes. Therefore, we are unable to provide a definitive answer based on direct experimental data.
4.Elaborating more in the future direction studies based on the results and discussion of this study can be done in the last section of discussion or even after conclusion. It can also clarify the boundaries that the studies. It also can held explaining that conclusions regarding the altitude (question/answer number 7) are based on the correlations and patterns observed in this study so that at some point the experiment may further prove this interactions.
Reply: We thank the reviewer for this valuable feedback. We have further elaborated on future research directions in the revised manuscript; please see lines 519-534,546-548 for specific details.